

# Cocoa colonic phenolic metabolites are related to HDL-cholesterol raising effects and methylxanthine metabolites and insoluble dietary fibre to anti-inflammatory and hypoglycemic effects in humans

Beatriz Sarriá, Miren Gomez-Juaristi, Sara Martínez López, Joaquín García Cordero, Laura Bravo and Mª Raquel Mateos Briz

Department of Metabolism and Nutrition, Institute of Food Science, Technology and Nutrition (ICTAN-CSIC); Spanish National Research Council (CSIC), Madrid, Spain

Corresponding authors
Beatriz Sarriá,
beasarria@ictan.csic.es
Mª Raquel Mateos Briz,
raquel.mateos@ictan.csic.es

## ABSTRACT

**Background:** In many cocoa intervention studies, health outcomes are related to cocoa components without taking into account the bioavailability of the main bioactive components: phenolic compounds and methylxanthines.

**Methods:** The present work associates the results of bioavailability and randomised controlled crossover studies in humans carried out with similar cocoa products, so that the main phenol and methylxanthine metabolites observed in plasma and urine are associated to the health effects observed in the chronic studies. We outstand that doses of cocoa and consumption rate used are realistic. In the bioavailability study, a conventional (CC) and a methylxanthine-polyphenol rich (MPC) cocoa product were used, whereas in the chronic study a dietary fibre-rich (DFC) and a polyphenol-rich (PC) product were studied in healthy and cardiovascular risk subjects.

**Results and Discussion:** The main phenolic metabolites formed after CC and MPC intake, 5-(4′-hydroxyphenyl)-γ-valerolactone-3′-sulfate, 3′-methyl-epicatechin-5-sulfate, 4-hydroxy-5-(4′-hydroxyphenyl)valeric acid-sulfate, 5-phenyl-γ-valerolactone–sulfate and 5-(4′-hydroxyphenyl)-γ-valerolactone-3′-glucuronide, may contribute to the changes in cholesterol (and indirectly HDL-cholesterol) observed after the regular intake of both DFC and PC, in healthy and cardiovascular risk subjects, whereas 7-methylxanthine (the main cocoa methylxanthine metabolite) and theobromine, together with its content in insoluble dietary fibre, may be responsible for the decrease of IL-1β and hypoglycemic effects observed with DFC. With both phenolic and methylxanthine metabolites a strong dose–response effect was observed.

**Conclusion:** After the regular consumption of both DFC and PC, positive changes were observed in volunteer's lipid profile, which may be related to the long-lasting presence of colonic phenolic metabolites in blood. In contrast, the anti-inflammatory and hypoglycemic effects were only observed with DFC, and these may be related to methylxanthine metabolites, and it is likely that insoluble dietary fibre may have also played a role.

# INTRODUCTION

Cocoa is consumed worldwide in different population groups being Europe the world's largest cocoa consuming area with a 40% of global consumption (*Food and Agricultural Organization of the United Nations, 2003*). In response to the demands of consumers, the food industry continues producing functional cocoa products enriched with bioactive components, such as dietary fibre, polyphenols or methylxanthines, or making products without fat, sugar or sweeteners (*World Cocoa Conference, 2018*).

A body of scientific evidence supports the beneficial cardiovascular health effects of cocoa consumption (*Arranz et al., 2013*; *Ellam & Williamson, 2013*; *Gómez-Juaristi et al., 2011*; *Hooper et al., 2012*). However, there are aspects in cocoa's health effects that need to be clarified. The aim of this work is to look into the effects of cocoa on cardiovascular health through associating the lipid lowering, anti-inflammatory and hypoglycemic effects observed in two chronic, randomized, controlled studies, with the phenolic and methylxanthine metabolites observed in bioavailability studies in humans with similar cocoa products. It is noteworthy that the cocoa products used in both the chronic and bioavailability studies, produced by the same cocoa manufacturer, had a similar cocoa matrix and realistic amounts of cocoa were consumed by the volunteers in both studies.

In many cocoa intervention studies, health outcomes are related to cocoa component intake without taking into account the bioavailability of the main bioactive components in the test food. The present work goes one step ahead and considers the bioavailability of two of the main bioactive components in cocoa products: the phenolic compounds, mainly flavanols epicatechin, catechin and procyanidins (*Gómez-Juaristi et al., 2019*) as well as methylxanthines, mainly theobromine followed by theophylline and caffeine (*Martínez-López et al., 2014a*). The main metabolites observed in plasma and urine are related to the health effects observed in two randomised controlled human studies carried out with cocoa products similar to those used in the bioavailability study, that is, a functional cocoa product rich in dietary fibre (*Sarriá et al., 2014*) and a cocoa product rich in polyphenols (*Martínez-López et al., 2014b*). As said, the cocoa products used in the mentioned studies were provided by a well-known Spanish cocoa product manufacturer who prepared individual sachets containing the dose recommended on the label to make the chocolate drink. It is important to note that the doses used are realistic, contrary to many studies that use exaggerated cocoa conditions. Moreover, the consumption rate corresponds to a real consumption pattern, two sachets per day.

# MATERIALS AND METHODS

## Characterisation of the cocoa products

Five cocoa powders were analyzed, an un-processed raw cocoa powder (cocoa RC), and four new soluble cocoa products commercially available in Spain: cocoa DFC (rich in

dietary fibre (DF)), cocoa MPC (rich in methylxanthines and cocoa), cocoa CC (conventional cocoa low in sugar (2.8%)), and cocoa PC (rich in cocoa, and thus phenolic compounds, and low in sugar (2.8%)).

In all the cocoa products studied, polyphenols and methylxanthines were extracted following a procedure developed by our group (*Bravo & Saura-Calixto, 1998*). Total polyphenols were determined using Folin-Ciocalteau reagent and gallic acid as standard and the phenolic and methylxanthine composition of the cocoa extracts was characterized by high-performance liquid chromatography (HPLC) with diode-array detection (DAD) using an Agilent 1200 series equipment, the procedure followed is described in *Gómez-Juaristi et al. (2019)*.

The total DF of the cocoa products was analyzed in triplicate from defatted samples following the AOAC method modified in our laboratory (*Saura-Calixto et al., 2000*) and is described in *Sarriá et al. (2012a)*.

The antioxidant capacity of all the cocoa products was evaluated in the soluble extracts by three different methods. The reducing power of the samples was measured using the ferric reducing/antioxidant power (FRAP) assay (*Pulido, Bravo & Saura-Calixto, 2000*). The capacity of samples to scavenge the stable radical ABTS was determined by the ABTS discoloration method (*Re et al., 1999*), and the oxygen radical absorbance capacity (ORAC) was determined according to (*Huang et al., 2002*). In the three parameters, Trolox was used as standard and results were expressed as μmol of Trolox Equivalent (TE) per gram of dry matter (d.m.) of the product. Additionally, the FRAP and ABTS methods were also used to determine the antioxidant capacity in serum samples obtained in the bioavailability study (see below), and results were expressed as μM TE.

## Methylxanthine and polyphenol bioavailability studies

A crossover, single-blind study was carried out in healthy men and women, aged 18–45 years old, with body mass index between 18 and 25 kg/m$^2$. They were also non-smoker, non-vegetarian, non-pregnant women, who were not taking any medication or nutritional supplements, not suffering from any chronic pathology or gastrointestinal disorder. Fourteen volunteers gave their written informed consent prior to participation in the bioavailability study, but only thirteen completed the study.

On two different days, separated by a 10-day period, after an overnight fast, volunteers consumed 15 g of the conventional cocoa product low in sugar (CC) and 25 g of a cocoa product enriched in methylxanthines and polyphenols (MPC) in 200 mL of semi-skimmed milk. A nurse inserted a cannula in the cubital vein of one of their arms and blood samples were collected into EDTA-coated tubes at baseline ($t = 0$) and 0.5, 1, 2, 3, 4, 6 and 8 h after consuming the cocoa drinks and plasma was separated by centrifugation. Analytical methods were optimized to measure by chromatography the absorption of methylxanthines and phenolics in plasma. The two days previous to each intervention, certain polyphenol-rich-foods, such as some fruits, vegetables and their derivate beverages, as well as methylxanthine-rich-foods, such as coffee, tea or chocolate products, were restricted from their diets in order to reduce inter and intraindividual differences. Volunteers were asked to complete a 24 h food intake recall

the day before each intervention in order to control any possible food restriction incompliance. The bioavailability studies of polyphenols and methylxanthines in the cocoa products have already been described in detail in *Gómez-Juaristi et al. (2019)* and *Martínez-López et al. (2014a)*, respectively.

## Chronic cocoa studies

To evaluate the effects of regularly consuming DFC (*Sarriá et al., 2014*) and PC (*Martínez-López et al., 2014b*) on markers of cardiovascular health, two controlled, randomized, crossover studies were carried out in free-living healthy and moderately hypercholesterolemic (2.000–2.400 mg/L; 5.172–6.206 mmol/L) volunteers. In both studies the inclusion criteria was: men and women (not including pregnant women), between 18 and 55 years old, with body mass index (BMI) <30 kg/m$^2$, non-vegetarian, non-smoker, not suffering from any chronic pathology or gastrointestinal disorder. Briefly, each study consisted in a run-in (two weeks), a control and cocoa intervention (four weeks each). In *Sarriá et al. (2014)* volunteers consumed twice a day 15 g of DFC (30 g/day) and in *Martínez-López et al. (2014b)*, two times a day 7.5 g per serving of PC (15 g/day). Along both studies other cocoa products, certain fruits and vegetables rich in polyphenols were restricted. Forty four subjects completed both studies, among these were the 13 subjects who carried out the bioavailability study.

Blood samples, blood pressure, heart rate and anthropometric measurements were taken at baseline and at the end of each intervention. In blood samples, biochemical, inflammatory, oxidation and antioxidant biomarkers were measured, for more details please see *Sarriá et al. (2014)* and *Martínez-López et al. (2014b)*.

Both chronic and bioavailability studies were conducted according to the guidelines laid down in the Declaration of Helsinki and all procedures were approved by the Clinical Research Ethics Committee of Hospital Universitario Puerta de Hierro Majadahonda in Madrid (Spain). Approvals were obtained by this Hospital because all human studies carried out in Consejo Superior de Investigaciones Cientifícias (CSIC) are submitted to the Ethics Committee of Hospital Puerta de Hierro, unless there are participants in the project who are affiliated to a hospital with an Ethics Committee, but in the present study this was not the case. Volunteer recruitment was carried out through placing advertisements in the Universidad Complutense campus as well as through giving short talks between lectures.

## RESULTS AND DISCUSSION

In the recent years, a Spanish cocoa manufacturer has been producing new soluble, powder, cocoa products with added value increasing the content in bioactive compounds, specifically flavanols, dietary fibre and methylxanthines, and lowering the sugar levels trying not to compromise the flavour and texture of the cocoa products but to keep or improve health effects.

Certainly, in cocoa and most cocoa products, the main bioactive components are phenolic compounds, methylxanthines and dietary fibre. Therefore, phenolic compound and methylxanthine bioavailability studies were carried out in humans, using realistic

 

**Table 1** Flavanols antioxidant activity dietary fibre and methylxanthines composition of the soluble cocoa products used in bioavailability and chronic interventions in humans: cocoa rich in cocoa (PC), cocoa rich in dietary fibre (DFC), conventional cocoa (CC), cocoa rich in methylxanthines and cocoa (MPC) and raw un-processed cocoa (RC).

|  | Product PC | Product DFC | Product CC | Product MPC | Product RC |
|---|---|---|---|---|---|
| Polyphenol composition |  |  |  |  |  |
| Total polyphenols (a) (μg equiv gallic acid/g product) $n = 6$ | 34.04 ± 2.28 | 15.75 ± 0.67 | 21.70 ± 1.40 | 25.63 ± 1.00 | 42.11 ± 2.50 |
| EC (mg/g dry matter) (b) | 1.26 ± 0.18 | 0.33 ± 0.09 | 0.57 ± 0.07 | 1.15 ±006 | 2.40 ± 0.50 |
| CA (mg/g dry matter) (b) | 0.47 ± 0.03 | 0.26 ± 0.12 | 0.32 ± 0.03 | 0.53 ± 0.04 | 0.83 ± 0.17 |
| PB1 (mg/g dry matter) (b) | 0.20 ± 0.04 | n.d. | 0.04 ± 0.02 | 0.23 ± 0.02 | 0.41 ± 0.11 |
| PB2 (mg/g dry matter) (b) | 1.09 ± 0.10 | 0.57 ± 0.11 | 0.39 ± 0.05 | 0.82 ± 0.06 | 2.04 ± 0.68 |
| Total flavanols (b) | 3.02 ± 0.35 | 1.16 ± 0.32 | 1.32 ± 0.17 | 2.73 ± 0.18 | 5.68 ± 1.46 |
| Methylxanthines composition (b) |  |  |  |  |  |
| Theobromine (mg/g dry matter) (b) | 6.43 ± 0.84 | 5.11 ± 0.14 | 5.63 ± 0.06 | 7.08 ± 0.22 | 6.33 ± 0.17 |
| Theophylline (mg/g dry matter) (b) | 0.01 ± 0.01 | n.d. | n.d. | 0.13 ± 0.02 | 0.06 ± 0.01 |
| Caffeine (mg/g dry matter) (b) | 0.88 ± 0.08 | 0.51 ± 0.05 | 0.66 ± 0.06 | 3.03 ± 0.28 | 1.46 ± 0.21 |
| Total methylxanthines (b) | 7.32 ± 0.93 | 5.62 ± 0.19 | 6.29 ± 0.12 | 10.24 ± 0.52 | 7.85 ± 0.39 |
| Dietary fibre composition (c) |  |  |  |  |  |
| Soluble dietary fibre (%) = NS + UA (c) | 3.13 ± 0.59 | 1.68 ± 0.13 | 2.69 ± 0.70 | 3.00 ± 0.86 | 4.27 ± 0.27 |
| Neutral sugars (%) (c) | 2.46 ± 0.43 | 0.69 ± 0.04 | 2.21 ± 0.51 | 1.80 ± 0.67 | 2.70 ± 0.14 |
| Uronic acid (%) (c) | 0.67 ± 0.16 | 0.99 ± 0.09 | 0.48 ± 0.19 | 1.20 ± 0.19 | 1.57 ± 0.13 |
| Insoluble dietary fibre (%) = NS + UA (c) | 11.96 ± 1.05 | 20.32 ± 1.67 | 14.31 ± 0.51 | 26.63 ± 2. 54 | 11.58 ± 1.30 |
| Neutral sugars (%) (c) | 10.49 ± 0.96 | 19.06 ± 1.60 | 13.32 ± 0.29 | 25.43 ± 2.39 | 9.12 ± 1.08 |
| Uronic acid (%) (c) | 1.47 ± 0.09 | 1.26 ± 0.07 | 0.99 ± 0.22 | 1.20 ± 0.15 | 2.46 ± 0.22 |
| Total dietary fibre (%) (c) | 15.09 ± 1.64 | 22.00 ± 1.80 | 17.00 ± 1.21 | 29.63 ± 3.40 | 15.85 ± 1.57 |

**Note:**
Analysis carried out using (a) Folin-Ciocalteau method (b) HPLC method (c) *Bravo & Saura-Calixto (1998)*. n.d., not detected; EC, epicatechin; CA, catechin; PB1, procyanidin B1; PB2, procyanidin B2.

amounts of cocoa products from the same cocoa manufacturer with the purpose to better understand the metabolism of these components and to establish a relationship between the main cocoa phenolic and methylxanthine metabolites (*Gómez-Juaristi et al., 2019*; *Martínez-López et al., 2014a*) and the biological effects observed in chronic studies (*Sarriá et al., 2014*; *Martínez-López et al., 2014b*).

## Characterization of cocoa products

The polyphenolic composition, according to the Folin-Ciocalteau assay and HPLC-DAD analysis, the methylxanthine and dietary fibre composition are shown in Table 1. Cocoa powder RC showed the highest total polyphenol content which may be explained because product RC is naturally rich, non-processed cocoa. Among the commercialized cocoa rich products, cocoa powder PC showed the highest level of total polyphenols followed by cocoa products MPC and CC, whereas product DFC, which was rich in DF but not in cocoa, showed the lowest polyphenol content. According to the results obtained in cocoa RC and DFC, it seems that the manufacturing process lowers the content of polyphenols approximately 62.6% and from this point cocoa enrichment has taken

**Table 2** Antioxidant activity of cocoa products rich in cocoa rich in cocoa (PC), cocoa rich in dietary fibre (DFC), conventional cocoa (CC), cocoa rich in methylxanthines and cocoa (MPC) and raw un-processed cocoa (RC).

| | Product PC | Product DFC | Product CC | Product MPC | Product RC |
|---|---|---|---|---|---|
| Antioxidant activity | | | | | |
| FRAP (µmol TE/µg product) $n = 3$ | 175.07 ± 1.52 | 75.88 ± 2.61 | 109.41 ± 2.33 | 120.12 ± 3.26 | 219.33 ± 6.90 |
| ABTS (µmol TE/g product) $n = 3$ | 133.09 ± 3.70 | 56.64 ± 3.26 | 66.64 ± 6.88 | 89.53 ± 2.99 | 173.42 ± 7.53 |
| ORAC (µmol TE/g product) $n = 8$ | 459.71 ± 9.40 | 234.51 ± 4.89 | 248.56 ± 4.32 | 257.39 ± 5.79 | 483.93 ± 19.88 |

place to different extents to reach the polyphenol levels described in products PC, CC and MPC. Chromatographic analyses of the cocoa products studied showed that, among the monomeric flavanols, epicatechin (EC) was the most abundant, between 0.33 and 2.40 mg/g d.m., whereas catechin (CA) ranged between 0.26 and 0.83mg/g d.m. and, among the dimeric compounds the most abundant was procyanidin B2 (PB2), 0.39–2.04 mg/g d.m., whereas procyanidin B1 (PB1) was between 0 and 0.41 mg/g d.m. It is noteworthy that the "natural" content of total polyphenols obtained in product RC is not reached by any of the cocoa enriched products. EC and CA contents observed in the natural cocoa powder were similar to those described by *Miller et al. (2009)* in other cocoa products.

Since many years ago, it has been established that food antioxidant capacity is strongly related to its polyphenol content (*Benzie & Szeto, 1999*). In accordance, in the present work the phenolic content of the studied cocoa powders was directly proportional to the antioxidant activity assessed using FRAP, ABTS and ORAC assays (Table 2), so that FRAP results were five times higher than the total polyphenol content of the cocoa products, whereas ABTS and ORAC results were 3–4 and 10–14 times higher, respectively. Taking the present results into account, consuming a dose of 15 g/d of cocoa PC, as recommended by the manufacturer, would yield 2,626 (FRAP method), 1,996 (ABTS method) or 3,855 (ORAC method) µmol Trolox equivalents (TE) of antioxidant capacity, exceeding that provided by a 100 mL serving of coffee (2,267 (FRAP) and 1,328 (ABTS) µmol TE, respectively) or red wine (601 (FRAP) and 631 (ABTS) µmol TE, respectively), which are the beverages that have been described to have the highest antioxidant capacity within the Mediterranean diet by *Saura-Calixto & Goñi (2006)*.

Furthermore, soluble cocoa products are also a relevant source of DF, in contrast to chocolate, due to DF being largely discarded in its production (*Jenkins et al., 2000*). Dietary guidelines recommend a minimum daily intake of DF of 25 g (https://health.gov/dietaryguidelines/2015/), which is considerably higher than the estimated intakes in Western countries (*Jenkins et al., 2000*). Bearing this in mind, the food industry prompted the production of dietary fibre rich foods, such as cocoa product DFC. In all products, insoluble DF (IDF) was the predominant fraction, varying between 73% and 92% of the total DF content, which is in agreement with previous results (*Lecumberri et al., 2007*) that showed that the main polysaccharides in cocoa IDF were

cellulose, hemicellulose and some pectic substances, while cocoa soluble DF (SDF) was composed mainly of pectins and minor amounts of galactomannans.

As for the different cocoa products, it is noteworthy that cocoa product MPC, which is rich in polyphenols and methylxanthines, showed the highest content in total DF (30%), even higher than the fibre-rich cocoa product DFC, which contained 22% total DF. Both products DFC (92%) and MPC (90%) presented a similar proportion of IDF with respect to total DF (Table 1). The contribution of SDF to the total DF was highest in product RC (27%) followed by product PC (20.7%). Interestingly, SDF content in the cocoa products followed the same order than the antioxidant capacity and their total flavanol content (R > PC > MPC > CC > DFC).

Regarding the methylxanthine contents in cocoa, theobromine was the primary methylxanthine in all the tested cocoa products (Table 1), followed by caffeine, in agreement with the literature (*Menguy et al., 2009*). Cocoa MPC was the product with the highest content in total methylxanthines, being particularly rich in caffeine, as this product was enriched in kola nut, which is a source of caffeine. Theophylline was only found in products MPC and RC, and scarcely in PC, whereas in the other cocoa products the concentrations were under the limit of detection. Cocoa is also a source of caffeine. A 15 g serving of cocoa MPC would provide 153.6 mg of methylxanthines (69% theobromine and 29.5% caffeine), which is similar to the amount of caffeine provided per serving in 330 mL of a cola drink (40 mg).

## Bioavailability and metabolism

It is essential to better understand the bioavailability and metabolism of the main bioactive compounds in cocoa, that is, polyphenols. However, another question that needs to be further clarified is what other compounds in cocoa, beyond flavanols, are responsible for the beneficial health effects. In this context, methylxanthines constitute another outstanding group of bioactive compounds in cocoa products, although less considered when biological effects of cocoa products on health are discussed. For this reason, it is relevant to explore the main metabolites formed from methylxanthines' bioavailability along with the metabolites derived from metabolism of flavanols, as it can help to better explain the compounds responsible for the health effects associated to cocoa consumption and the involved mechanisms.

In the bioavailability studies presented in this work, a main objective was, besides establishing flavanol and methylxanthine absorption and metabolism after intake of realistic doses of cocoa products, to assess if there was a dose-dependent response. Therefore, the product which showed the highest amount of methylxanthines of all the cocoa products and which was rich in polyphenols (MPC) was selected to compare with the conventional product (CC).

## Methylxanthines

The absorption and metabolism of methylxanthines present in the two tested soluble cocoa products, CC corresponding to the conventional cocoa low in sugar and MPC enriched in methylxanthines and polyphenols, were evaluated in healthy subjects
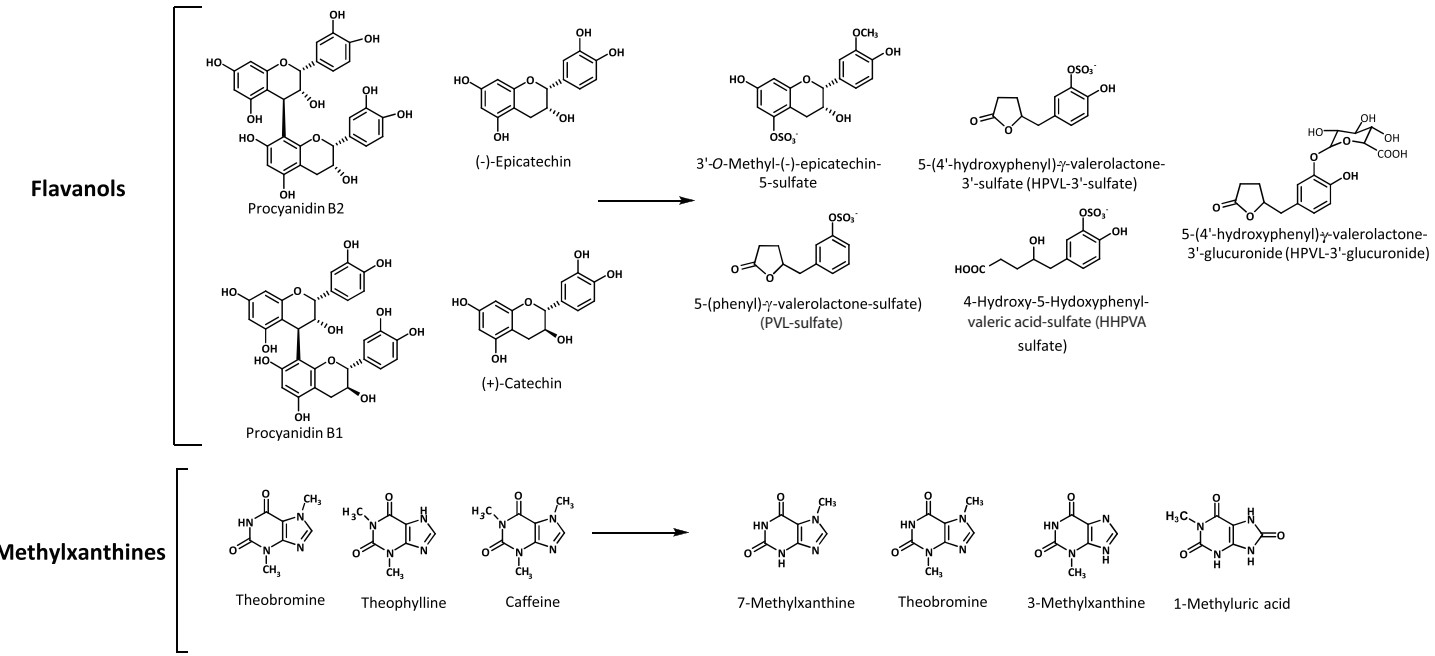

**Figure 1** **Bioactive compounds present in the cocoa products and the most abundant flavanol and methylxanthine metabolites formed after cocoa intake.**

(*Martínez-López et al., 2014a*). The main methylxanthines and their plasma and urine metabolites were characterized after consuming realistic doses of cocoa products (15 g of CC and 25 g of MPC, in 200 mL of milk) with different methylxanthine composition: CC provided 84.45 mg of theobromine (TB) and 9.9 mg of caffeine (CF) while MPC provided 177 mg of TB, 75.75 mg of CF and 3.25 mg of theophylline (TP).

Metabolism of cocoa methylxanthines in humans has not been widely studied and some results obtained by our group were novel findings. Up to 12 different metabolites were identified and quantified in a single run in biological fluids by HPLC-DAD and LC-MS-QTOF. On one hand, TB, CF, TP and paraxanthine (PX) together with two monomethylxanthines (3-methylxanthine (MX) and 7-methylxanthine) were identified and quantified in plasma by HPLC-DAD. TB was the most abundant plasmatic compound of all the identified metabolites, followed by TP and PX with similar levels. These compounds were rapidly detected in plasma, showing their maximum concentration ($C_{max}$) between 1 and 4 h after ingestion and maintaining similar levels throughout 8 h, not returning to basal levels in this period. The maximum concentration ($C_{max}$) and the area under curve (AUC) values of all metabolites were significantly higher after the intake of MPC, showing a dose-dependent response.

These metabolites together with 1-methylxanthine and different mono-, di- and trimethyluric (MU) acids (1-MU, 1,3-MU, 1,7-MU, 3,7-MU and 1,3,7-MU) involved in the biotransformation of the ingested methylxanthines were detected by LC-MS-QTOF and quantified by LC-DAD in urine, with 7-MX as the most abundant metabolite followed by TB and 3-MX (Fig. 1). Most metabolites were extensively excreted in urine between 4 and 8 h after CC and MPC intake; although similar to the behaviour described in plasma,

higher levels of excreted metabolites were observed until 24 h post-intake. Results showed a strong dose–response effect after cocoa consumption at the doses tested, up to 256 mg of MX. It can be concluded that methylxanthines in cocoa are bioavailable, partially metabolized and rapidly eliminated, with a sustained urinary excretion for as long as 24 h after intake. The long permanence of methylxanthines and their metabolites in body fluids favours the involvement of these compounds on the health effects associated to cocoa products consumption.

## Polyphenols

Although there are many bioavailability studies of cocoa flavanols, most have been carried out with unrealistic doses (*Ottaviani et al., 2012a*, *2012b*; among others). Addressing the need to carry out studies that follow realistic consumption guidelines to further understand the bioavailability and metabolism of real doses of flavanols, which may differ from higher doses used in previous articles, a recent clinical intervention study was performed in our research group (*Gómez-Juaristi et al., 2019*). This study was aimed at assessing in healthy humans the absorption and metabolism of polyphenols in a single, realistic dose of CC (15 g) and MPC (25 g), which provided 19.8 mg and 68.25 mg of flavanols, respectively.

Blood and urine samples were taken before and after CC and MPC consumption and analyzed by LC–MS-QToF. Up to 10 and 30 metabolites were identified in plasma and urine, respectively. Among the metabolites identified, phase II derivatives of epicatechin showed the maximum concentration in plasma from 1 to 2 h after ingestion and kinetics compatible with absorption at small intestine. 3′-Methyl-epicatechin-5-sulfate and epicatechin-3′-sulfate were the predominant metabolites followed by 4′-methyl-epicatechin-5-sulfate, and epicatechin-3′-glucuronide among others. Phase II derivatives of epicatechin accounted for approximately 33% of total urinary metabolites. The most abundant group of metabolites were phase II derivatives of hydroxyphenyl-γ-valerolatones and valeric acid, formed at colonic level as 5C-ring fission metabolites by microbiota-mediated biotransformation (Fig. 1). These compounds showed delayed kinetics due to their colonic origin and represented the most abundant group of metabolites (67% of total urinary metabolites) headed by 5-(4′-hydroxyphenyl)-γ-valerolactone-3′-sulfate (HPVL-3′-sulfate) followed by 4-hydroxy-5-(hydroxyphenyl)valeric acid sulfate (HHPVA-sulfate), 5-(phenyl)-γ-valerolactone—sulfate (PVL—sulfate), 5-(4′-hydroxyphenyl)-γ-valerolactone-3′-glucuronide (HPVL-3′-glucuronide) and 5-(3′,4′-dihydroxyphenyl)-γ-valerolactone (DHPVL), among others. According to these results, HPVL-3′-sulfate could be a sensitive biomarker of flavanol intake, considering its abundance in urine, amounting up to 40% of total urinary metabolites. Flavanols of CC and MPC showed a dose-dependent absorption with a recovery value of 35% of the ingested dose.

## The relationship between the main cocoa metabolites and the cardiovascular health effects

To understand the health effects of cocoa or cocoa-based functional products, it is important to carry out randomized controlled human studies using normal, realistic doses

of the food products. Such aspects were considered in two chronic studies carried out in our research group; in the first, the regular consumption of a cocoa product rich in dietary fibre (DFC) (Table 3) during four weeks led to a significant increase in HDL-cholesterol and a decrease in glucose and interleukin (IL)-1β without producing changes in body weight neither other anthropometric parameters studied (*Sarriá et al., 2014*); in the second, the sustained consumption of the cocoa product rich in cocoa (PC) (Table 3) for the same length of time, four weeks, resulted in an increase in HDL-cholesterol without other changes in cardiovascular related biomarkers (*Martínez-López et al., 2014b*). When the health effects of both cocoa products were comparatively studied in relation to the amounts of bioactive compounds consumed, without considering the bioavailability of these compounds, on one hand, mainly the flavanol content provided by both cocoa products was related to the increase in HDL-cholesterol, and on the other hand, the IDF and theobromine were associated to the hypoglycemic and anti-inflammatory effects observed after consuming the fibre rich product (*Sarriá et al., 2015*). However, as aforementioned, the present study goes one step ahead and further looks into the relationship between the main phenolic and methylxanthine metabolites observed in bioavailability studies in humans, and the cardiovascular effects observed in the chronic intervention studies.

## Main phenolic metabolites after cocoa consumption in humans and their health effects

### Epicatechin and phase II derivatives

Numerous human dietary intervention studies link the consumption of flavan-3-ols derived from cocoa to improved cardiovascular health (*Heiss, Keen & Kelm, 2010*; *Rodriguez-Mateos et al., 2014*), having epicatechin and its phase II derivatives, at least partially, been linked with these beneficial effects (*Borges et al., 2018*). In contrast to the results of *Dower et al. (2015)*, who investigated the effects of pure epicatechin supplementation and described no effects on cholesterol, many other studies have correlated the consumption of products rich in epicatechin with a decrease in serum levels of LDL-cholesterol in both hypercholesterolemic subjects (*Grassi et al., 2005*) and healthy subjects (*Baba et al., 2007*). The mechanisms responsible for the decrease in the levels of LDL include: inhibiting cholesterol absorption in the gut, reducing cholesterol synthesis by decreasing the activity and/or expression of hydroxymethylglutaryl coenzyme A (HMG-CoA) synthase, HMG-CoA reductase, sterol *O*-acyltransferase and microsomal triglyceride transport protein in the liver, suppressing hepatic secretion of apolipoprotein B100, increasing the expression of hepatic LDL receptors, and preventing oxidation of LDL (*Baba et al., 2007*). Separately, epicatechin and catechin, which have similar biological activity, have also shown to raise the concentration of HDL-cholesterol through the increased expression of scavenger receptor B type I (SR-BI), sterol regulatory element binding proteins (SREBPs), ATP binding cassette transporter A1 (ABCA1) or apolipoprotein A1, among others (*Martínez-López et al., 2014b*). Moreover, *Rodríguez-Mateos et al. (2018)* recently described that cocoa consumption in healthy

**Table 3 Summary of the characteristics and outcomes of the cocoa bioavailability and chronic studies.**

| Reference | Characteristics of the study | Number of volunteers and criteria | Cocoa product and dose | Main cocoa metabolites* | Health effects |
|---|---|---|---|---|---|
| Gómez-Juaristi et al. (2019) | Polyphenol bioavailability study Randomized and crossover, 8 h long carried out on two separated days | 13 Healthy subjects | Day 1: 15 g of conventional cocoa (CC)—19.80 mg of flavanols/day Day 2: 25 g of cocoa rich in methylxanthines and phenols (MPC)—68.25 mg of flavanols/day | Plasma: epicatechin and its phase II derivatives: 3′-Methyl-epicatechin-5-sulfate, epicatechin-3′-sulfate, 4′-methyl-epicatechin-5-sulfate, epicatechin-3′-glucuronide, among others. 5-(3′,4′-Dihydroxyphenyl)-γ-valerolactone (DHPVL) and its phase II derivatives (HPVL-3′-glucuronide, HPVL-3′-sulfate, PVL-methyl-glucuronide). Urine: phase II derivatives of epicatechin, approximately 33% of total. The majority were phase II derivatives of 5-(4′-hydroxyphenyl)-γ-valerolactone (HPVL) and 4-hydroxy-5-(4′-hydroxyphenyl)valeric acid (HHPVA): HPVL-3′-sulfate followed by HHPVA-sulfate, PVL-sulfate and HPVL-3′-glucuronide, among others Dose-dependent response | No changes in blood pressure along the 8 h. No changes in antioxidant capacity in blood along the 8 h, except 1 h (ABTS) and 2 and 8 h (FRAP) |
| Martínez-López et al. (2014a) | Methylxanthine bioavailability study Randomized and crossover, 8 h long carried out on two separated days | 13 Healthy subjects | Day 1: 15 g of conventional cocoa (CC)—84.45 mg of theobromine (TB) and 9.9 mg of caffeine (CF)/day Day 2: 25 g of cocoa rich in methylxanthines and phenols (MPC)—177 mg TB, 75.75 mg CF and 3.25 mg theophylline (TP)/day | Plasma: TB, CF, TP and paraxanthine (PX) together with two monomethylxanthines (3-methylxanthine (MX) and 7-MX) Urine: TB, CF, TP and PX, along with monomethylxanthines (1-MX, 3-MX and 7-MX) and different mono-, di- and tri- methyluric (MU) acids (1-MU, 1,3-MU, 1,7-MU, 3,7-MU and 1,3,7-MU). 7-MX as the most abundant metabolite followed by TB and 3-MX Dose-dependent response | No changes in blood pressure along the 8 h. No changes in antioxidant capacity in blood along the 8 h, except 1 h (ABTS) and 2 and 8 h (FRAP) |
| Sarriá et al. (2014) | Randomized, controlled and crossover chronic study. Cocoa and control interventions were four weeks long | 44 healthy subjects (n = 24) and moderately hypercholesterolemic (n = 20) | 30 g/day (2 doses of 15 g of cocoa rich in dietary fibre (DFC)) 34.8 mg of flavanols/day 153.3 mg of TB and 15.3 mg of CF/day 6.6 g total dietary fibre (DF) 0.5 g soluble DF/day 6.1 g insoluble DF/day | Phenol metabolites according to Gómez-Juaristi et al. (2019): Epicatechin and phase II derivatives of epicatechin Phase II derivatives of hydroxyphenyl-γ-valerolactones and valeric acid Methylxanthine metabolites according to Martínez-López et al. (2014a): TB, CF, TP and paraxanthine (PX) together with monomethylxanthines (1-MX, 3-MX and 7-MX) and mono-, di- and tri-MU acids (1-MU, 1,3-MU, 1,7-MU, 3,7-MU and 1,3,7-MU) Dietary fibre according to Sarriá et al. (2014). | ↑HDL-cholesterol ↓Blood glucose ↑IL-1β ↓IL-10 |

| Reference | Characteristics of the study | Number of volunteers and criteria | Cocoa product and dose | Main cocoa metabolites[*] | Health effects |
|---|---|---|---|---|---|
| *Martínez-López et al. (2014b)* | Randomized, controlled and crossover-chronic study. Cocoa and control interventions were four weeks long | 44 healthy subjects (*n* = 24) and moderately hypercholesterolemic (*n* = 20) | 15 g/day (two doses of 7.5 g) of cocoa rich in cocoa (PC) 45.3 mg of flavanols/day 96.45 mg of TB and 13.2 mg of CF/day 2.3 g total dietary fibre (DF) 0.5 g soluble DF/day 1.8 g insoluble DF/day | Phenol metabolites according to *Gómez-Juaristi et al. (2019)*: Epicatechin and phase II derivatives of epicatechin Phase II derivatives of hydroxyphenyl-γ-valerolactones and valeric acid Methylxanthine metabolites according to *Martínez-López et al. (2014a)*: TB, CF, TP and PX together with monomethylxanthines and different mono-, di- and tri-MU acids Dietary | ↑HDL-cholesterol |

Note:
[*] 5-(3′,4′-dihydroxyphenyl)-γ-valerolactone (DHPVL); 5-(4′-hydroxyphenyl)-γ-valerolactone (HPVL); 5-phenyl-γ-valerolactone (PVL); 4-hydroxy-5-(3′,4′-dihydroxyphenyl)valeric acid (HDHPVA) and 4-hydroxy-5-(hydroxyphenyl)valeric acid (HHPVA).

humans induced effects on flow mediated dilatation, pulse wave velocity and blood pressure mediated by EC metabolites and not DHPVL metabolites.

In addition, it is important to emphasize epicatechin's antioxidant function, being able to adhere to LDL particles or apolipoprotein B and to recycle molecules of α-tocopherol donating a hydrogen atom, thus maintaining the concentration of endogenous antioxidants for longer (*Wan et al., 2001*). This property, combined with the reduction of iron (non-heme) and malondialdehyde (MDA) concentrations and the suppression of pro-oxidant enzymes, have been described to be responsible for the increase in blood antioxidant activity after epicatechin consumption (*Prakash, Basavaraj & Murthy, 2019*). However, not all phase II metabolites have the same antioxidant activity. Epicatechin and its 7-*O*-glucuronide derivative show a similar reduction in LDL oxidation, unlike 3′-*O*-glucuronide and 4′-*O*-methyl-3′-*O*-glucuronide derivatives, which show lower activity. Similarly, the *O*-methylated derivatives C-3′ and C-4′-*O*-methyl ether of epicatechin are less active than their base compound. Moreover, it should not be disregarded that the antioxidant activity of these metabolites is dependent on the pH of the medium, so that at 7.4 their activity is largely retained, which confirms their antioxidant action in physiological conditions (*Monagas et al., 2010*).

There is controversy regarding the hypoglycaemic activity of epicatechin and its derivatives. According to *Rodríguez-Mateos et al. (2018)*, there were no effects of epicatechin or procyanidins on fasting glucose concentration. In contrast, a study conducted by *Josic et al. (2010)* in healthy subjects showed that consumption of green tea rich in epicatechin reduced glucose and insulin concentration. The main pathway by which epicatechin and its derivatives exert this effect is through increased sensitivity to insulin in hepatic or adipose cells, in which epicatechin inhibits the action of different signalling proteins such as PKC, IKK, JNK, NF-κB and PTP-1B, which leads to a decrease in insulin resistance (*Cremonini et al., 2016*). In addition, a recent study in mice has

observed that the consumption of a food rich in epicatechin promotes the regeneration of cells β-pancreatic (*Prakash, Basavaraj & Murthy, 2019*).

### 5-(3′,4′-Dihydroxyphenyl)-γ-valerolactone (DHPVL)

5-(3′,4′-Dihydroxyphenyl)-γ-valerolactone (DHPVL) is a compound resulting from the degradation of procyanidins or monomers that may reach the colon and be cleaved by the colonic microbiota (*Appeldoorn et al., 2009*; *Fogliano et al., 2011*). Its antioxidant capacity has been tested in vitro, being superior to catechin, ascorbic acid and Trolox (vitamin E analogue; *Appeldoorn et al., 2009*).

Up to date, hardly any in vivo study has been conducted to demonstrate the effect of DHPVL on lipid and glycaemic metabolism. Only by extrapolating the results obtained in clinical studies with products rich in procyanidins, such as cocoa (*Sarriá et al., 2014*; *Martínez-López et al., 2014b*; *Sarriá et al., 2015*; *Grassi et al., 2005*; *Baba et al., 2007*; *Khan et al., 2012*), with its antioxidant capacity (*Khan et al., 2012*), and considering the fact that it is one of the main metabolites found in blood after the consumption of cocoa (*Urpi-Sarda et al., 2009a*; *Urpi-Sarda et al., 2009b*) it has been deduced that DHPVL has hypolipemic and hypoglycemic properties. This is in agreement with the results obtained by *Rodríguez-Mateos et al. (2018)*, who described that the consumption of cocoa procyanidins induced a health benefit in healthy humans related to the reduction of total cholesterol, having an impact on cholesterol absorption and faecal steroid excretion inside the gastrointestinal tract. This effect was particularly linked to procyanidins content, which are metabolized into DHPVL and its phase II derivatives.

Recently, a study carried out in rabbits with obesity and non-alcoholic fatty liver suggested that the effects of procyanidins, metabolized into DHPVL derivatives, were due to changes in the intestinal microbiota. In fact, the supplementation with procyanidin B2 decreased the ratio *Firmicutes/Bacteroidetes* and increased the proportion of *Akkermansia*, which has the ability to maintain the thickness of intestinal mucus, reducing the permeability to lipopolysaccharides (LPS) and thus relieving inflammation. As a consequence of the maintenance of the intestinal barrier, a decrease in the concentration of LPS in serum was observed, which may explain the reduction in the accumulation of triglycerides in the liver and the protection against non-alcoholic fatty liver (*Xing et al., 2019*).

According to the results on phenolic metabolites (Table 3), it may be proposed that the main metabolites observed after intake of CC and MPC (HPVL-3′-sulfate, 3′-methyl-epicatechin-5-sulfate, HHPVA-sulfate, PVL—sulfate and HPVL-3′-glucuronide), may be responsible for certain changes observed in the chronic studies with DFC and PC. Specifically, to the changes in HDL-cholesterol observed after regular intake of DFC or PC in both healthy and cardiovascular risk subjects.

Regarding the effects of the cocoa products on serum antioxidant capacity, along the bioavailability study, there were no significant differences between the two cocoa products according to the estimations carried out with FRAP method except at 2 h ($p = 0.039$) and 6 h ($p = 0.044$), and according to ABTS at 1 h ($p = 0.004$) with MPC higher than CC (Table 4). In agreement, no significant differences in serum antioxidant capacity, measured

**Table 4 Antioxidant capacity in serum samples along the bioavailability study using ABTS and FRAP methods.**

| Hours | 0 | 0.5 | 1 | 2 | 3 | 4 | 6 | 8 |
|---|---|---|---|---|---|---|---|---|
| **ABTS (µM TE)** | | | | | | | | |
| CC | 3,830.59 ± 114.35 | 3,682.88 ± 112.79 | 3,605.06 ± 81.59[b] | 3,658.54 ± 94.03 | 3,696.70 ± 91.70 | 3,665.45 ± 97.03 | 3,687.98 ± 82.26 | 3,797.14 ± 108.21 |
| MPC | 3,944.78 ± 62.59 | 3,913.31 ± 65.85 | 3,953.47 ± 49.67[a] | 3,860.47 ± 61.00 | 3,831.18 ± 58.11 | 3,873.56 ± 63.21 | 3,828.77 ± 45.97 | 3,815.37 ± 73.11 |
| | N.S. | N.S. | $p = 0.004$ | N.S. | N.S. | N.S. | N.S. | N.S. |
| **FRAP (µM TE)** | | | | | | | | |
| CC | 592.82 ± 28.14 | 579.43 ± 32.27 | 551.75 ± 24.63 | 540.19 ± 26.67 | 550.39 ± 24.61 | 550.49 ± 27.44 | 556.48 ± 31.65 | 530.39 ± 24.16 |
| MPC | 596.70 ± 23.77 | 597.63 ± 22.09 | 580.28 ± 22.50 | 583.84 ± 23.20 | 578.61 ± 23.43 | 566.25 ± 26.51 | 583.85 ± 26.35 | 560.42 ± 28.83 |
| | N.S. | N.S. | N.S. | $p = 0.039$ | N.S. | N.S. | N.S. | $p = 0.044$ |

**Note:**
Conventional cocoa (CC), cocoa rich in methylxanthines and cocoa (MPC), Trolox Equivalents (TE). Values with different superscript letters (a,b) are significantly different.

by ORAC, FRAP and ABTS as well as levels of protein (carbonyl groups) and lipid (MDA), were observed in the chronic interventions (*Sarriá et al., 2014*; *Martínez-López et al., 2014b*).

## Main methylxanthine metabolites after cocoa consumption in humans and their health effects

All the methylxanthine compounds have in common the capacity to act as phosphodiesterase inhibitors, modulators of GABA and adenosine receptors, as well as to regulate intracellular calcium levels (*Williams et al., 1978*; *Monteiro et al., 2016*). However, *Williams et al. (1978)* described many years ago that there are certain differences in the inhibitory and regulatory activity, with theophylline and 3-methylxanthine having greater activity than caffeine and 1-methylxanthine.

Traditionally, the HDL cholesterol raising effect of cocoa has been associated with cocoa flavanols (*Baba et al., 2007*; *Mellor et al., 2010*; *Neufingerl et al., 2013*). Nevertheless, methylxanthines, particularly theobromine, may increase HDL-concentration in blood (*Neufingerl et al., 2013*), having been postulated that the mechanism of action involves increased levels of apolipoprotein A-1 and is independent of its activity as an adenosine receptor inhibitor (*Monteiro et al., 2016*, *2019*). It is also important to highlight the properties of methylxanthines in counteracting hyperglycaemia and insulin resistance. This hypoglycemic effect is explained by the ability of these compounds to regulate intracellular levels of cAMP, with the release of insulin by β-pancreatic cells and glucose by the liver being dependent on this second messenger (*Monteiro et al., 2016*).

To end, the anti-inflammatory properties of methylxanthines should also be emphasized. While caffeine is able to inhibit the expression of tumour necrosis factor (TNF)-α by suppressing the cyclic-adenosine monophosphate/protein kinase A (cAMP/PKA) pathway and inhibiting cAMP phosphodiesterase (*Horrigan, Kelly & Connor, 2004*) no variation in the production of IL-1β, IL-12 and IL-10 has been observed in human studies (*Horrigan, Connor & Kelly, 2004*; *Loftfield et al., 2015*; *Haskó & Cronstein, 2011*). In the case of theobromine, a decrease in IL-1β levels was reported in studies conducted in vitro (*Fuggetta et al., 2019*) and in mice (*Camps-Bossacoma et al.,*

*2019*). In contrast, in humans, no studies with theobromine and inflammatory markers other than C-reactive protein (CRP) were found. However, two studies, one carried out in moderate hypercholesterolemic and in healthy subjects (*Sarriá et al., 2014*), observed a decrease in the serum levels of IL-1β and IL-10 after consumption of a cocoa product rich in fibre, so we cannot rule out a synergistic effect between the two bioactive compounds (*Goya et al., 2016*).

According to these results on methylxanthine metabolites, 7-methylxanthine (the main cocoa methylxanthine metabolite) and theobromine may be responsible for the antiinflammatory (decrease of IL-1β) and hypoglycemic effects observed after the intervention with DFC.

## Dietary fibre

The remaining bioactive compound in cocoa products, dietary fibre, has certainly played a role on the cardiovascular related effects observed. The different intake of dietary fibre, with DFC (over 10 g/d) and PC (less than 4 g/d) may be related to the decrease in glucose ($p = 0.029$) levels observed only after regularly consuming DFC, but not PC (*Sarriá et al., 2015*). This result is in agreement with the significant decrease in plasma glucose concentration ($p = 0.019$) observed in a previous study carried out with a dietary fibre-rich cocoa product (*Sarriá et al., 2012b*), where the mechanisms responsible for this effect were suggested to be that dietary fibre rich foods delay glucose absorption from the small intestine (*Giacco et al., 2000*) and improve insulin sensitivity (*Tokede, Gaziano & Djoussé, 2011*), since it is well known that dietary fibre rich foods delay glucose absorption from the small intestine and improve insulin sensitivity (*Sarriá et al., 2015*). However, the higher dietary fibre intake with DFC did not increase HDL-cholesterol, in contrast to results described by *Jenkins et al. (2000)* who reported a significant increase in HDL-cholesterol after consumption a cocoa-bran (25 g DF/d) for two weeks in healthy subjects.

The effects of DFC on inflammation may be related to the down-regulation of cytokine gene expression and up-regulation of the expression of *scgb1a1*, gene that codifies a protein related to anti-inflammation, in the colon tissue (*Massot-Cladera et al., 2017*), as observed in rats that consumed either a cocoa diet or a cocoa fiber diet. In addition to the positive effects on inflammation and glucose, DFC also produced positive gastrointestinal effects such as an increased number of daily bowel movements and reduced time to have a bowel movement, without inducing major adverse gastrointestinal symptoms (*Sarriá et al., 2012a*).

This study presents the following limitations and strengths: the bioavailability and chronic studies were carried out separately, using different cocoa products although they were produced by the same manufacturer and the cocoa matrix was quite similar. However, it is relevant that health outcomes of chronic cocoa consumption in humans are related to the cocoa's composition taking into account the bioavailability of the main bioactive components, also evaluated in a human study. As aforementioned, no previous studies, to our knowledge, have considered this integrative and associative approach. Both the chronic and bioavailability studies are well designed and they have been carried out in an adequate number of subjects with similar characteristics using realistic

amounts of cocoa. Moreover, the present work has been accomplished knowing that a dose-dependent response occurs with both polyphenols and methylxanthines, when the cocoa products were consumed at realistic doses, and the metabolites obtained are the same disregarding the different quantities of the bioactive components in cocoa products studied. It is pertinent to further investigate the results here presented, so that in future chronic cocoa studies in humans a bioavailability study will be performed simultaneously.

## CONCLUSIONS

After the regular consumption of both DFC and PC, positive changes were observed in volunteer's lipid profile, which may be related to the long-lasting presence of colonic phenolic metabolites (headed by 5-(4′-hydroxyphenyl)-γ-valerolactone-3′-sulfate) in blood. In contrast, the anti-inflammatory and hypoglycemic effects were only observed with DFC, and these may be related to methylxanthine metabolites, particularly 7-methylxanthine which was the main cocoa methylxanthine metabolite, and theobromine, and it is likely that insoluble dietary fibre may have also played a role.

## ABBREVIATION

| | |
|---|---|
| BMI | Body mass index |
| CF | Caffeine |
| CA | Catechin |
| CC | Conventional cocoa |
| DF | Dietary fibre |
| DFC | Dietary fibre-rich cocoa |
| EC | Epicatechin |
| IDF | Insoluble DF |
| MDA | Malondialdehyde |
| MU | Methyluric acid |
| MPC | Methylxanthine-polyphenol rich cocoa |
| PX | Paraxanthine |
| PC | Polyphenol-rich cocoa |
| PB1 | Procyanidin B1 |
| PB2 | Procyanidin B2 |
| RC | Raw cocoa |
| SDF | Soluble DF |
| TB | Theobromine |
| TP | Theophylline |
| TE | Trolox equivalent |
| DHPVL | 5-(3′,4′-Dihydroxyphenyl)-γ-valerolactone |
| HHPVA-sulfate | 4-Hydroxy-5-(Hydroxyphenyl)valeric acid sulfate |
| HPVL-3′-sulfate | 5-(4′-Hydroxyphenyl)-γ-valerolactone-3′-sulfate |

| HPVL-3′-glucuronide | 5-(4′-Hydroxyphenyl)-γ-valerolactone-3′-glucuronide |
| PVL-sulfate | 5-(Phenyl)-γ-valerolactone-sulfate |

## ACKNOWLEDGEMENTS

We are grateful to the volunteers who participated in the study.

### Funding

Nutrexpa, S.L. funded the study and the Spanish Ministry of Economy and Competitivity (MINECO-FEDER), projects AGL2010-18269 and AGL2015-69986-R provided financial support. Sara Martínez López was a predoctoral fellow of the JAE Program (JAEPre097) co-funded by CSIC and the European Social Fund. Miren Gómez-Juaristi was a predoctoral FPI fellow BES2008-007138 of the Spanish Ministry of Science and Innovation. The funders had no role in study design, data collection and analysis, decision to publish, or preparation of the manuscript.

### Grant Disclosures

The following grant information was disclosed by the authors:
Spanish Ministry of Economy and Competitivity (MINECO-FEDER): AGL2010-18269 and AGL2015-69986-R.
JAE Program: JAEPre097.
CSIC and European Social Fund.
Spanish Ministry of Science and Innovation: BES2008-007138.

### Competing Interests

The authors declare that they have no competing interests.

### Author Contributions

- Beatriz Sarriá conceived and designed the experiments, performed the experiments, analyzed the data, prepared figures and/or tables, authored or reviewed drafts of the paper, and approved the final draft.
- Miren Gomez-Juaristi performed the experiments, analyzed the data, prepared figures and/or tables, and approved the final draft.
- Sara Martínez López performed the experiments, analyzed the data, prepared figures and/or tables, and approved the final draft.
- Joaquín García Cordero analyzed the data, prepared figures and/or tables, and approved the final draft.
- Laura Bravo conceived and designed the experiments, authored or reviewed drafts of the paper, and approved the final draft.
- Mª Raquel Mateos Briz conceived and designed the experiments, performed the experiments, analyzed the data, authored or reviewed drafts of the paper, and approved the final draft.

## Human Ethics

The following information was supplied relating to ethical approvals (i.e., approving body and any reference numbers):

Both chronic and bioavailability studies were approved by the Clinical Research Ethics Committee of Hospital Universitario Puerta de Hierro Majadahonda in Madrid (Spain).

## Data Availability

The raw measurements are available in the Supplemental Files.

## Supplemental Information

Supplemental information for this article can be found online at http://dx.doi.org/10.7717/peerj.9953#supplemental-information.

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
