# Peer review of "Cocoa colonic phenolic metabolites are related to HDL-cholesterol raising effects and methylxanthine metabolites and insoluble dietary fibre to anti-inflammatory and hypoglycemic effects in humans"

_PeerJ, doi:10.7717/peerj.9953_

## Round 0.1 · original submission · Major Revisions

A figure and/or table with integrated results from different studies could benefit the manuscript. The aim of the study should be clearly indicated in the introduction section of the manuscript. For being considered as a "Research Articles", the title of the manuscript should be changed according to the study and results. As it is now, this sounds like a "Review Article".

Reviewer 1 ·

Basic reporting

This is an interesting and well written paper. They relation the bioavailability of cocoa polyphenols and they CV properties.

Experimental design

The experimental design of the paper is not clear.
The informmation given in the title doesn't correspond with the aim of the paper.
Authors should characterize better the fibre content of cocoa, because the decrease of glucose levels and reduce the inflammation are associated to fibre group. So, in 3.2 section they should include an study characterizing the fibre content.

Validity of the findings

The results obtained should be integrated in a better way. To start, I strongly reccomend to incorporate a figure, or table describing the intervention studies, population, dose, main metabolites, health effects...

Indeed, authors should compare the difference in the polyphenol metabolism between single and chronic intake, it should be interesting including a venn diagram to analyze the common and different metabolites...

If epicatechin induce a decrease in LDL cholesterol, why the authors doesn't have similar results, as they say, the cocoa supplementation increases the HDL cholesterol, but... They also should analyze the functionality of HDL to appreciate the real effect of cocoa.

I considered very interesting if the author's could analyze the antioxidant capacity of the patient's serum

Additional comments

1. FAO reference is missing in the bibliography.
2. In Results& discussion authors loose the
3. I miss a list of abbreviation. There are too many abbreviation very similar which difficult the follow of the paper.
4. L. 223 respectively
5. L 119 trolox equivalent (TE); again L 220 Trolox equivalent (TE)
6. The affirmation between the relationship between the AOX in vitro capacity and in vivo capacity today is very controversial...
7. L 279 mg per how many quantity of starting material?the dose tested in the human experiment?
8. Table 3--> why the authors present the results in blood and urine together? And also, why they only choose one main compound of each family?
9. l 395, 396 397... revise glucuronide word.
10. L 418 reference?
11. L 442 where is the table supporting this??
12. How many hours separate the last cocoa intake to blood collection?
13. table 1: legend of EC, CA, PB1...
13. Supporting informmation: is not described in the text; also, Table 1 and table 2 (excel files) are written in spanish language.
Also; table cocoa bioavailability and pharmacokinetic the colors are not described which study correspond.

Reviewer 2 ·

Basic reporting

This article on “exploring remaining secrets behind the bioavailability of cocoa bioactive compounds and cardiovascular health effects in humans” is an interesting article that combines several complementary study designs to explore these remaining secrets. The article is well written but could benefit from a figure presenting the different study designs included in this study. Also an overview figure presenting the integrated results from all different studies would add extra value to the article. There is a lot of interesting results presented in this article but is a bit scattered throughout the article; as such a summary figure could be beneficial for the reader.

Experimental design

no comments

Validity of the findings

It would be good to elaborate a bit more on the strengths and limitations section just before the conclusion. The multi-tiered approach including complementary study designs can be included as a strength for instance.

Additional comments

Minor comments:

Abstract

Results: “Whereas, 7-methylxanthine (the main cocoa methylxanthine metabolite) and theobromine may be responsible for the decrease of IL-1β and hypoglicemic effects observed after the intervention with DFC, in addition to its content in insoluble dietary fibre.”, this sentence seems not final, part seems missing?

Conclusion:
• hypoglicemic effects: spelling correction “hypoglycaemic effects”
• “… and it likely that insoluble dietary fibre may have also played a role.” The “is” seems missing between it and likely.


Page 14:
“Therefore, the product which showed the highest amount of methylxanthines of all the cocoa products and rich in polyphenols (MPC) was selected to compare with the conventional product (CC).”; suggestion to add “and which was” in front of “rich in polyphenols”.

Page 11:
“lowers the content of polyphenols approximately 62.6%”; suggestion to add “with” in front of approximately.

Page 14:
“…metabolites were characterized after consuming realistic doses cocoa of products with”; suggestion for rephrasing: “…metabolites were characterized after consuming realistic doses of cocoa via products with”

Page 14:
“…between 1 to 4 hours and maintaining similar levels throughout 8 h”; suggestion for rephrasing: “…between 1 to 4 hours after ingestion and maintaining similar levels throughout 8 h”

Page 16:
“…showed the maximum concentration in plasma from 1 to 2 h and kinetics compatible with their absorption at small intestine.”; suggestion for rephrasing: “…showed the maximum concentration in plasma from 1 to 2 h after ingestion and kinetics compatible with their absorption at the small intestine.”

Page 16:
Conversion problem of character Ɣ on line 326: “hydroxyphenyl-฀-valerolatones”

Throughout text you sometimes write “Attending to these results”, I believe you mean “According to these results”?

Page 18:
“The main pathway by which epicatechin and its derivatives exert this effect is through increased sensitivity to insulin in hepatic or adipose cells, in which epicatechin acts inhibiting the action of different signalling proteins such as …”; suggestion for rephrasing: “The main pathway by which epicatechin and its derivatives exert this effect is through increased sensitivity to insulin in hepatic or adipose cells, in which epicatechin inhibits the action of different signalling proteins such as …”

Page 20:
“Attending to the results on phenolic metabolites, it may be concluded the main metabolites observed …” this sentence is far too long and hard to read, maybe you can split it in 2 sentences to improve clarity and readability? I would also suggest to replace the “Attending to” by “According to”.

Page 22:

The following sentence is unclear: “However, we carried out the current work knowing that a dose-dependent response occurs with both polyphenols and methylxanthines, when cocoa products were consumed at realistic doses, and no matter the quantity of these compounds, the metabolites obtained are the same.”; Not sure what you mean with “and no matter the quantity of these compounds, the metabolites obtained are the same” for instance?

Reviewer 3 ·

Basic reporting

The manuscript has been clearly written with professional English and sufficient bibliographic references. Tables, figures and raw data have been shared.
Regarding the list of references, write them in order that they appear in the main text and not in numbers (guide for authors).

Experimental design

Although the experimenal design has been well defined, with suffient detail and authors have performed rigorous clinical investigation, authors are trying to use previous published works and data related to chronic cocoa studies what is a great limitation to compare the bioavailability and chronic studies because thay were carried out separately and used different cocoa products.

Validity of the findings

The bioavailability study provides novel results. However, the objective of the work is not clear when authors use different cocoa products, that have been used in the bioavailability study, to evaluate the chronic effects on cardiovascular health. Moreover conclusions related to health effects asociated to bioactive compounds of cocoa can not be linked to original research presented in this work because bioavailability and chronic studies are not comparable.

Additional comments

Authors should focus on bioailability study of cocoa compounds (phenolic compounds and methylxanthines) that provide novelty results and should consider to provide new finding related to the chronic heath effects of these compounds in a better designed clinical trial.

---

## Round 0.2 · accepted · Accept

The authors have revised the manuscript according to the reviewer comments.